# Short Bouts of Physical Activity Are Associated with Reduced Smoking Withdrawal Symptoms, But Perceptions of Intensity May Be the Key

**DOI:** 10.3390/healthcare8040425

**Published:** 2020-10-23

**Authors:** Marianna Masiero, Helen Keyworth, Gabriella Pravettoni, Mark Cropley, Alexis Bailey

**Affiliations:** 1Applied Research Division for Cognitive and Psychological Science, European Institute of Oncology IRCCS, 20141 Milan, Italy; gabriella.pravettoni@unimi.it; 2Department of Biomedical and Clinical Sciences, University of Milan, 20157 Milan, Italy; 3School of Biosciences & Medicine, Faculty of Health and Medical Sciences, University of Surrey, Surrey GU2 7XH, UK; helen.keyworth@gmail.com; 4Department of Oncology and Hemato-Oncology, University of Milan, 20122 Milan, Italy; 5School of Psychology, Faculty of Health and Medical Sciences, University of Surrey, Surrey GU2 7YH, UK; mark.cropley@surrey.ac.uk; 6Institute of Medical and Biomedical Education, St George’s University of London, London SW17 0RE, UK

**Keywords:** cigarette smoking, affect regulation, moderate exercise, withdrawal, decision-making, personality

## Abstract

The primary aim of this study was to assess the effectiveness of a short bout (10 min) of moderate-intensity exercise to reduce withdrawal symptomatology, craving and negative affect; while the secondary aim was to assess how the effectiveness of a short bout of moderate exercise can be modulated by the perception of intensity in physically active and low-activity smokers. Fifty low-activity and physically active smokers were recruited (24 male and 26 female) and randomized in three different conditions. Prescribed (objective) moderate intensity (OBJ) and perceived moderate intensity (PER), and passive waiting (PW). After the intervention (T3), smokers reported less desire to smoke in the PER (*p* < 0.001) and OBJ (*p* < 0.001) conditions, relative to the PW condition. At T3 smokers in the PER condition reported less negative affect than smokers in the PW condition relative to the baseline (T1) (*p* < 0.007). Further, smokers in the PER condition reported less negative affect than smokers in the PW condition (*p* < 0.048). Physically active (PA) smokers perceived less exertion than low-activity (LA) smokers, and the effects were stronger in the PER condition relative to OBJ. Generally, our results suggest that a short bout of moderate exercise helps both LA and PA smokers. These findings provided a novel insight into the psychological mechanisms that affect the efficacy of the exercise in smoking cessation and suggest that exercise should be tailored according to individual perception of intensity.

## 1. Introduction

Tobacco cigarette smoking is the leading cause of premature death, accounting for more than seven million deaths each year [1,2]. Combined programs of clinical counselling and first-line (nicotine replacement treatments) and/or second-line treatments (bupropion and varenicline) are currently the more effective aids. Nevertheless, less than 30% of people successfully quit and relapse rates are high [3,4], with less than 10% remaining abstinent after six months [5,6,7]. The accruing evidence demonstrates that a short bout of moderate intensity exercise (only 10 min) is sufficient to reduce withdrawal symptoms [8], cravings [9], reward, and decrease the intention to smoke [10], in abstinent overnight cigarette smokers. Similar effects have been found in smokers trying to quit who engage in vigorous-intensity exercise [11].

Van Rensburg and colleagues (2013) observed that aerobic exercise was associated with increased time to light the next cigarette after a period of nicotine deprivation, as well as reducing subjective and neural response to cigarette cues [12,13]. Relaxing activities such as yoga (30 min), moderate walking (30 min) and cycling (10 min) are associated with improvements in positive affect and reductions in negative affect [3,14,15]. In particular, researchers observed that the enhancement of positive affect occurred both “during” and “following” exercise [3]. Similarly, De Jesus and Prapavessis (2018) observed that a bout of moderate intensity exercise in abstinent smokers produced a significant increase in positive affect (2.55 points versus 0.79), and a reduction in negative affect (2.55 points versus 0.29) compared with the control condition [16], stressing a double impact of exercise on both negative and positive affect [16]. Scientific evidence recognizes the pivotal role of exercise in smoking cessation, yet only 22% of smokers use physical activity as an aid to control their smoking, while 35% had used it during a previous quit attempt [15].

Although there is clear evidence demonstrating the benefits of exercise on reducing smoking cravings, it is less clear what the underlying biological and psychological mechanisms that promote active quitting are [17,18]. For example, at the biological level Keyworth and colleagues (2018) observed an interaction between exercise and nicotine on α7 nicotine acetylcholine receptor up-regulation in the CA2/3 hippocampal region of mice [17]. This up-regulation was associated with the elimination of somatic nicotine withdrawal symptoms [17,18,19], suggesting that this region-specific upregulation may at least partly underline the positive effect of exercise in reducing nicotine withdrawal symptoms. In addition, in a comprehensive review of the literature, Ussher and colleagues (2014) concluded that exercise had a direct effect on the central nervous system and neurobiological processes, with increasing beta-endorphin levels thought to increase the desire to smoke. Moreover, at the psychological level, a large number of studies reported that physical exercise supports smoking abstinence by means of emotional regulation and tension reduction [3]. For example, according to Nesbitt’s Paradox [20], the cigarette is able to respond to two different needs: stimulation to cope with boredom, and relaxation to reduce tension. Exercise is able to reduce stress levels, helping smokers to contain their desire for cigarettes, and to reduce tension [3,21,22], improving mood and psychological well-being, and supporting the activation of more adaptive coping strategies for emotional regulation [23,24,25]. More recently, Abrantes and colleagues (2017) argued that exercise has antidepressant and mood-enhancing effects in depressed smokers [26], mediated by the enjoyment of physical activity. However, the evidence for the psychological basis of exercise as a cessation aid is equivocal. For example, not all studies agree on the role of distraction as an effect caused by exercise [8,27]. Similarly, there is mixed evidence for the effectiveness of exercise to help smokers cope with nicotine withdrawal symptoms and maintain temporary abstinence, suggesting an individual variability that should be investigated and explained. At the same time, some authors point out that it is not clear which kinds of activity and which intensity is able to strongly support temporary smoking abstinence by reliving withdrawal symptoms [28]. Several authors [8,19,27] have emphasized the importance of conducting further laboratory and clinical studies in order to determine which factors augment or reduce the effects of physical activity. In particular, the kind of motivational factors involved in the adoption of regular physical exercise has not been well investigated. Autonomous motivation or internal regulation, might act to reduce nicotine withdrawal symptoms and related affective distress, favouring temporary abstinence [29,30]. Evidence shows that the type of exercise and the intensity have different levels of effectiveness [9,10,12,14,21,31], but, to our knowledge no studies have examined how the choice of activity type or intensity may moderate the effectiveness of physical activity as an intervention to reduce withdrawal symptoms.

Self-determination theory (SDT) [32] helps us to understand the motivational factors that might support health behaviour changes over time. SDT stresses the role of autonomy, competence, relatedness and determination in changing behaviours. In particular, health behaviour change is promoted when individuals are able to endorse the personal value related to health practice (autonomous motivation or internal regulation), they feel competent at the activity (effectiveness) and they feel understood and cared for by other people [33,34]. SDT emphasizes the role of autonomous motivation that encompasses intrinsic motivation (doing something for the pleasure associated with the behaviour itself), integrated regulation (doing something that is congruent with our values) and identified regulation (individual value related to the behaviour’s outcome) [33]. Supporting SDT, research has found that a strong autonomous motivation and competence predicted a better engagement in physical activity [35] and abstinence from cigarette smoking [36]. Thus, autonomous control over exercise participation, including the choice of activity type or intensity, may moderate the effectiveness of physical activity as an intervention to reduce nicotine withdrawal and affective-related distress, and to achieve abstinence both in the short and long term. Current lifestyle may also modulate the effect of exercise. Broadly, evidence shows that smokers are less physically active than nonsmokers or intermittent smokers, while regular physical activity is associated with nonsmoking [18]. Physically active smokers consume fewer cigarettes per day, and the level of physical activity is negatively correlated with the risk of being a heavy or chronic smoker [37]. Physically active smokers have been found to have a better self-efficacy for quitting due to them adopting a healthy lifestyle [38].

Utilizing this framework, the rationale for the current study was to add to our understanding of the motivational aspects—in particular, having autonomous control on the intensity of exercise—in modulating physical activity as an effective intervention for reducing nicotine withdrawal symptoms and affective related distress. Accordingly, there were two main aims. Firstly, we aimed to assess the effectiveness of a short bout (10 min) of moderate intensity exercise to reduce withdrawal symptomatology, craving and negative affect. In line with previous work, we predicted that a short bout of moderate intensity exercise (10 min), after three hours of nicotine abstinence, would be associated with reduced smoking withdrawal symptoms, cravings and negative affect during (10 min) and after the intervention (30 min). Our second aim was to assess how the effectiveness of a short bout of moderate exercise can be modulated by the perception of intensity in physically active and low-activity smokers. We reasoned that the effectiveness of a short bout of moderate intensity exercise on desire to smoke and cigarette withdrawal symptoms is modulated by the perception of exercise intensity. We hypothesized that smokers who engaged in a session of a moderate exercise, established according to what each participant perceived to be moderate-intensity exercise, would report lower cravings for desire to smoke and cigarette withdrawal symptoms compared with smokers who engaged in a prescribed moderate-intensity exercise. Furthermore, we hypothesized that low-activity smokers would perceive moderate-intensity exercise as more intense than physically active smokers, and we reasoned that low-activity smokers would perceive moderate-intensity exercise as more beneficial than prescribed moderate-intensity exercise, relative to physically active smokers. In particular, we hypothesized that low-activity smokers would report a bigger reduction in craving and desire to smoke, and a better mood enhancement during and following exercise than physically active smokers.

## 2. Materials and Methods

### 2.1. Participants

Participants were recruited through different channels, including posters, mailing lists, and via personal invitation and snowball sampling. The inclusion criteria were as follows: 18–35 years of age; smoking at least 10 cigarettes a day for at least two years; having an expired carbon monoxide (ECO) concentration of 10 parts per million (ppm). The exclusion criteria were as follows: not receiving any form of psychiatric or medical treatment; not pregnant. The average number of cigarettes smoked per day was 13.48 (SD = 5.34) and the number of years as smokers was 6.48 (SD = 2.72). The sample size was established according to statistics available in the literature [21,22] and using G*Power [39]. An adequate effect size (from d = 0.01 “small” d = 0.06 “medium” to d = 0.14 “large”) was established in terms of Cohen (1988) [40,41], with a power (1–β) of 0.8, and an α-value of 0.05 [31]. Power analysis implemented using G*Power [39] determined that a sample size of 18 for both the low-activity and physically active smokers was acceptable in order to reach power of 0.8 and an α-value of 0.05.

#### 2.1.1. Procedures

The study was conducted at the University of Surrey. Participants were recruited using the School of Psychology’s SONA recruitment system and posters emailed to all University of Surrey staff and students, as well as via personal invitation. The recruitment strategy was based on snowball sampling. A set of inclusion criteria were established as follows: 18–35 years of age, not receiving any form of psychiatric or medical treatment and not pregnant. Participants were also required to smoke at least 10 cigarettes a day for at least two years and have an expired carbon monoxide (ECO) concentration of 10 parts per million (ppm); smokers who consume ≥10 cigarettes a day are more likely to experience the cravings and withdrawal symptoms which make abstinence difficult [42].

Each participant received a detailed presentation of the project by a trained researcher and those wishing to participate read and signed a written consent form. Participants were volunteers and individuals could withdraw their consent at any point during the study. Participants were requested to maintain normal smoking behaviour during the days before the experiment and to cease smoking three hours before the experimental session.

We decided to establish three hours of abstinence according to the evidence that this time is sufficient to detect differences in individual responses [21,27,31]. Each participant was asked to wake up at 7 a.m. and to smoke as normal till 10 a.m., and then to stop for three hours (from 10 a.m. to 1 p.m.). At 10 a.m. and at 1 p.m., ECO was assessed.

Resting heart rate was measured before three hours of abstinence (pre-abstinence), after three hours of abstinence within 30 min of the start of the intervention (post-abstinence), during (every 30 s during each intervention), and post-intervention (10 min after the intervention); while ECO was measured before three hours of abstinence (pre-abstinence) and after three hours of abstinence within 30 min of the start of the intervention (post-abstinence).

Heart rate was assessed using a Polar RS300X heart rate monitor (Polar Electro Oy, Kempele, Finland) with a chest band, while ECO was measured using a Bedfont Micro Smokerlyzer (Bedfont Scientific Instruments Ltd., Maidstone, UK).

Participants in the study were categorized as either low-activity (LA) and physically active (PA). Participants were considered as LA if they did not engage in high-intensity physical activity three or more times a week for at least 20 min, or moderate-intensity physical activity at least five times a week for 30 min; whereas individuals were considered as physically active if they engaged in more physical activity per week than this.

This was a within-subject design, thus participants were assigned to complete all three conditions in a counterbalanced order:Prescribed (objective) moderate intensity exercise (OBJ): participants were required to exercise for 10 min on a Corival bicycle ergometer (Lode, B. V., Groningen, The Netherlands) at a prescribed moderate intensity determined by the Karvonen method (55% heart rate reserve) [43];Perceived moderate intensity exercise (PER): participants were required to exercise for 10 min on the same ergometer at a prescribed moderate intensity according to what each participant perceived to be moderate-intensity exercise (Borg RPE level 3) [44];Passive waiting (PW): participants sat passively with magazines to read for 10 min.

All conditions were performed in one laboratory session and the total time spent for each condition was 30 min, while the time spent from one condition to other was 15 min. According to the Borg Rating of Perceived Exertion (RPE) moderate-intensity exercise (e.g., a 30-min brisk walk or a 15-min run) is equated with a heart rate of 64–79 and an RPE of 3 out of 10 (from *Nothing at all* to *Very, very heavy*) [4,44]. At baseline, the following questionnaires were administered: Fagerstrӧm Test of Nicotine Dependence [45], Motives for Physical Activity Measures-Revised [46] and Motivation for Smoking Questionnaire [47]; as well as Mood and Physical Symptoms Scale [48]; and Positive and Negative Affect Scale [49]. Participants completed the Mood and Physical Symptoms Scale [48] and Positive and Negative Affect Scale [49] at 0, 5, 10, 15, and 20 min post-exercise. Finally, Rating of Perceived Exertion [4,44] was administered 2.5 min and 7.5 min into each 10 min of exercise. The University of Surrey Ethics Committee (EC/2010/FHMS&FAHS) approved this study, and the study was conducted in accordance with the Helsinki Declaration (59th WMA General Assembly, Seoul, 2008).

#### 2.1.2. Measures

Fagerstrӧm Test of Nicotine Dependence (FTND): A six-item self-administered questionnaire, assessing both physical and psychological dependence. It evaluates three main dimensions, including the average daily number of cigarettes smoked, nicotine compulsion, and the general level of dependence. The total score ranges from 0 to 10, where: 0–2, mild dependence; 3–4 not severe dependence; 5–6 strong dependence; 7–10 very strong dependence [45]. This measure has been found to be valid and reliable [46,50].

Motives for Physical Activity Measures-Revised (MPAM-R): A thirty-item self-administered questionnaire on a seven-point Likert scale (ranging from 1 = “Not at all true for me” to 7 = “Very true for me”). It assesses the main motives for participating in physical activities. It is organized into five subscales: fitness (five items, e.g., “Because I want to be physically fit”); appearance (six items, e.g., “Because I want to define my muscles so I look better”); competence/challenge (7 items, e.g., “Because I like activities which are physically challenging”); social (5 items, e.g., “Because I want to meet new people”); enjoyment (7 items, e.g., “Because I enjoy spending time with others doing this activity”) [46]. Ryan and colleagues reported that all subscales had a good validity and reliability, supported by Cronbach’s value ranging from 0.78 to 0.92 [51]. Similar results were supported by other studies [52,53,54,55].

Positive and Negative Affect Scale (PANAS): A twenty-item self-administered questionnaire that assesses positive and negative affect on a five-point Likert scale (ranging from 1 = “Very slightly or not at all” to 5 = “Extremely”). It can be used to assess mood on various time scales by altering the instructions. Possible time scales can include this moment (state) and past month or longer (trait), but it was used in the present study as a state measure [49]. Cranford and Henry (2004) supported the validity and reliability of the PANAS, reporting a Cronbach’s value ranging from 0.89 (positive affect) to 0.85 (negative affect) [54].

Rating of Perceived Exertion (RPE): A self-administered scale to assess individual effort, exertion, breathlessness, and fatigue during physical activity. Generally, it can be used to measure exertion and pain [4,44]. Several studies have supported its validity and reliability [55,56,57].

Mood and Physical Symptoms Scale (MPSS): An eight-item self-administered questionnaire on a seven-point Likert scale (ranging from 1 = “Not at all” to 7 = “Extremely”). The core items of the MPSS consist of five single-item ratings of depressed mood, irritability, restlessness, hunger, and poor concentration. These items allow for assessment of nicotine withdrawal. It also includes two items on a seven-point Likert scale (ranging from 1 = “Strongly disagree” to 7 = “Strongly agree”) that assess the desire to smoke [48,58,59]. This measure has been found to be valid and reliable [60,61].

Motivation for Smoking Questionnaire (SMQ)*:* A twenty-seven-item self-administered questionnaire using a five-point Likert scale (ranging from 1 = “Not at all” to 5 = “Very much”) that assesses self-reported motives relating to features of abstinence. Individuals rate the following motivations: coping with stress; socialization; boredom; concentration; reducing discomfort related to being abstinent; monitoring weight; increasing enjoyment [47].

#### 2.1.3. Data analysis

Descriptive statistics are used to depict characteristics of the sample (see Table 1). A one-way ANOVA was implemented to assess differences between PA and LA smokers (factor) for gender, age, dependence level, number of years as a smoker, timing of the last cigarette smoked before the experiment, ECO pre- and post-abstinence, heart rate pre- and post-abstinence (dependent variables). A Student’s *t*-test was performed to assess differences between PA and LA smokers (factor) and motivation to be engaged in physical activity according to the MPAM-R (dependent variable). A series of repeated measures ANOVAs with mixed designs (within-subjects variables and between subjects variables) were implemented to assess the effect of three different conditions (perceived moderate exercise, prescribed moderate exercise and passive waiting) on withdrawal symptoms, and positive and negative affect at all timepoints (pre-intervention Time 1–baseline; Time 2–10 min of exercise, and Time 3–30 min post-intervention), according to participants’ status (PA or and LA). When Mauchly’s test of sphericity indicated that the assumption of sphericity was violated, Greenhouse and Geisser and Huynth-Feldt corrections were used according to the ε value. When appropriate, Bonferroni post hoc tests were applied. Data were analyzed using SPSS (IBM, USA) version 23.0.

## 3. Results

### 3.1. Characteristics of the Sample

A one-way ANOVA reported no significant differences between PA and LA smokers (see Table 1), except for the number of years as smokers (F(1) = 5.458, *p* < 0.024), with PA smokers smoking less (M = 5.58, SD = 2.215) compared with the LA smokers (M = 7.31, SD = 2.923) (Table 1).

### 3.2. Motives for Physical Activity Measure Revised (MPAM-R)

The statistic of Student’s *t-*test was used to assess differences between PA and LA smokers and motivation for engaging in physical activity. Status (PA, LA) was treated as an independent variable, while the five subscales of MPAM-R (interest, competence, appearance, fitness and social) and total motivation for engaging in physical activity were dependent variables. The following subscales were statistically significant: interest, (t(48) = −3.575, *p* < 0.001); competence, (t(48) = −2.151 *p* < 0.003); appearance, (t(48) = −2.751, *p* < 0.008); fitness, (t(48) = −3.121, *p* < 0.003); and total motivation to be engaged in physical activity, (t(48) = −3.555, *p* < 0.001). The social subscale was not significant (Table 2). As reported in Table 2, both PA and LA smokers identified “Interest” as their main motivational factor.

### 3.3. Motivation for Smoking Questionnaire (SMQ)

The statistic of Student’s *t-*test was used to assess differences in motivation for smoking between PA and LA smokers. No differences were observed between PA and LA smokers.

### 3.4. Desire to Smoke

For ‘Desire to smoke’, there was a statistically significant main effect of time (F(2) = 44.228, *p* < 0.001, η^2^ = 0.48), and condition (F(1.571, 75.407) = 25.347, *p* < 0.001, η^2^ = 0.346), and a condition by time interaction (F(3.038, 145.811) = 17.324, *p* < 0.001, η^2^ = 0.265). Bonferroni post hoc tests revealed significant differences between PER and PW conditions at Time 3 (*p* < 0.001); between OBJ and PW conditions at Time 2 and Time 3 (*p* < 0.001).

Overall, smokers in PER (*p* < 0.001) and OBJ (*p* < 0.001) reported less desire to smoke than smokers in PW at Time 3 (Figure 1). Smokers in OBJ reported less desire to smoke at Time 2. As shown in Table 3, the desire to smoke at Time 3 increased more for smokers in OBJ than in PER. No effects of status were found.

### 3.5. Withdrawal Symptoms

The eight items of the MPSS where highly correlated, and therefore a new variable named “Total Withdrawal Symptoms” was calculated. There was a statistically significant main effect of time (F(2) = 27.224, *p* < 0.001, η^2^ = 0.362), and condition (F(1.582, 73.334) = 5.784, *p* < 0.009, η^2^ = 0.108), and a significant condition by time interaction (F(2.720,130.566) = 6.553, *p* < 0.001, η^2^ = 0.120). There was no effect of status. Bonferroni post hoc tests revealed differences between PER and PW at Time 2 (*p* < 0.028) and Time 3 (*p* < 0.021); between OBJ and PW at Time 2 (*p* < 0.021) and Time 3 (*p* < 0.017). In particular, a significant reduction in total withdrawal symptoms was found at Time 2 for smokers in PER and OBJ, while at Time 3, withdrawal symptoms increased both in PW and OBJ (Figure 2).

### 3.6. Positive Affect (PANAS)

For positive affect, there was a statistically significant effect of time (F(1.617, 77.601) = 32.001, *p* < 0.001, η^2^ = 0.39), and condition (F(1.513, 72.633) = 5.312, *p* < 0.013, η^2^ = 0.077), and a significant condition by time interaction (F(2.915, 139.9) = 4.932, *p* < 0.001, η^2^ = 0.083). No effects due to status were found. Bonferroni post hoc tests revealed differences between PER and PW conditions (*p* < 0.007), and between OBJ and PW conditions (*p* < 0.001) at Time 2, reporting an increase in positive emotions (Figure 3).

### 3.7. Negative Affect

For negative affect there was a statistically significant main effect of time (F(1.661, 79.711) = 5.586, *p* < 0.008, η^2^ = 0.104) and condition (F(2) = 3.534, *p* < 0.048, η^2^ = 0.061), but there was no significant condition by time interaction. No effects of status were found. Bonferroni post hoc tests revealed a difference between Time 1 and Time 3 (*p* < 0.007) in the PER condition. At Time, smokers in PER reported less negative affect than smokers in PW at Time 1 (Figure 4). Considering the total amount of negative emotion, smokers in PER reported less negative emotion than smokers in PW (*p* < 0.048).

### 3.8. Rating of Perceived Exertion

Rating of perceived exertion was measured 2.5 min (Time 1) and 7.5 min (Time 2) into each 10 min of intervention. A repeated measures ANOVA was implemented using condition (PER, OBJ, PW) and time (Time 1 and Time 2) as within-subject variables and physical status as between subject variables. There was a statistically significant effect of condition (F(1.674, 80.364) = 315.590, *p* < 0.001, η^2^ = 0.868), and time (F(1) = 19.485, *p* < 0.000, η^2^ = 0.289), and a significant condition by time interaction (F(1.740, 83.500) = 5.232, *p* < 0.010). Bonferroni post hoc tests revealed that OBJ and PER conditions reported more perceived of exertion at T1 (*p* < 0.001) and T2 (*p* < 0.001) than PW. Smokers in PER reported less perceived exertion than smokers in OBJ at Time 1 (*p* < 0.001) and at Time 2 (*p* < 0.001) (Table 3).

There was also a significant difference between PA and LA smokers (F(1) = 9.168, *p* < 0.004, η^2^ = 0.160). LA smokers reported higher exertion than PA both at Time 1 and Time 2. Overall, PA smokers perceived less exertion than LA, and this was particularly evident in the PER condition relative to the OBJ condition, with the LA smokers in PER reporting less exertion both at Time 1 and Time 2 compared with smokers in OBJ (*p* < 0.002).

## 4. Discussion

Our findings revealed that a short bout of moderate intensity exercise reduced cigarette withdrawal symptoms and increased positive affect in temporarily abstinent smokers. The current study provides a novel insight and extends the previous literature of the psychological mechanisms underlying the efficacy of exercise as a strategy for reducing nicotine withdrawal symptoms during temporary abstinence [8,9,10]. In particular, this study provides insight about the role of autonomous motivation/internal regulation in the modulation of withdrawal symptoms and affective-related distress during physical exercise.

Our first hypothesis was partially supported by the results. Smokers that performed exercise that was perceived as moderate and smokers who were prescribed exercise reported lower strength of the desire to smoke than smokers in the passive waiting condition. Smokers in OBJ reported an increase in the desire to smoke at Time 3 compared with smokers in PER, while both conditions supported the reduction in desire to smoke and craving symptoms at Time 2 (during the exercise session).

Overall, the desire to smoke was reduced during exercise and increased thirty minutes following exercise. This finding does not support previous research [62]; indeed, in PW the desire to smoke and withdrawal symptoms exceeded the baseline values, suggesting a general worsening in nicotine withdrawal symptoms and desire to smoke. This opposite trend should be treated with caution, as it might be partially due to different baseline values observed both for desire to smoke and withdrawal symptoms. It is important to stress that smokers in all three groups (OBJ, SED and PW) were not attempting to quit. Therefore, after they had completed the required tasks the forthcoming “*availability of smoking*” may have increased their desire to smoke.

In addition, exercise was found to have a key role in affect modulation. Mainly, smokers in the PER and OBJ conditions reported greater positive affect than smokers in PW during and following exercise. This supported our theory that exercise can be used by smokers as a coping strategy and effective emotional regulation. Positive affect increased during exercise and declined following exercise. Crucially, smokers in the PER condition reported less negative affect than smokers in the OBJ and PW conditions. This result is also in accordance with Nesbitt’s Paradox [20], which highlighted the effect of exercise in reducing negative emotion, but also added an important theoretical insight. The reduction in negative emotions (e.g., distressed, irritability, hostility) is facilitated when exercise is perceived as moderate instead of prescribed.

Elibero and colleagues (2011) [14] argue that physical activity may help smokers to reduce affective motivation for cigarettes and could be used as a coping strategy. The present results reinforce the notion that physical activity helps smokers to cope with the negative effects of nicotine deprivation (e.g., irritability, difficult concentrating, depression, restlessness) and acts as an affect regulation strategy. Similarly, we argue that the positive affect induced by physical activity may foster individual motivation to maintain abstinence.

Contrary to expectations, there were no differences between PA and LA smokers. For example, Reed and colleagues (1998) reported that exercise is effective for smokers with an active lifestyle [63]. As expected, however, PA smokers perceived less exertion than LA smokers, and the effects were stronger in the PER condition relative to OBJ. In particular, LA smokers in the PER condition perceived less exertion both during and following exercise. Generally, our results suggest that a short bout of moderate exercise helps both LA and PA smokers. We argue that the different outcomes in the effectiveness of the exercise reported in smokers might be attributed to other factors such as the perception of intensity and the motivation to engage in physical activity Moreover, we argue that allowing smokers to choose the intensity of the exercise might be more beneficial in helping them to achieve the abstinence and to quit than providing specific exercise intensity guidelines.

According to SDT [34], autonomous control over exercise participation may mediate its effectiveness. In particular, the perception of control might help tobacco cigarette smokers to increase their intrinsic motivation to engage in physical exercise. Thus, smokers that freely choose the intensity of exercise believe they are able to set targets that are in line with their personal needs and goals. This may explain why smokers in the PER condition achieved greater control of their withdrawal symptoms and levels of negative affect. When individuals are more autonomously motivated, they experience more drive to achieve health outcomes [64]. Furthermore, the modulation of affect may play a key role in maintaining abstinence. In particular, the increase in positive affect may have moderated the relationship between exercise and withdrawal symptoms.

## 5. Limitations

The study presents some limitations that constrain the generalizability of our results. First, the study was conducted in a laboratory setting under controlled conditions, and the smokers in this study were relatively young and healthy, so we do not know if the findings transfer to the real world or to older individuals. Consistent with this point, the younger age of participants connected to the sampling strategy used might have caused a selection bias; this raises issues of ecological validity. It is therefore important to implement simultaneous studies outside of the lab to assess the effects of physical activity in modulating the desire to smoke and withdrawal symptomatology in daily life with a range of individuals. Furthermore, participants in the PW condition did not perform specific mental exercises of moderate intensity, for example, answering a quiz, responding to a questionnaire, or reading a page of a book and then answering some questions. We suppose that this might have increased the risk of incurring a distraction effect that might have influenced the emotional and behavioural activation of the participants. The mean age of participants was 18–35 years old. Therefore, participants covered two stages of Arnett’s classifications (2000) [65]: emerging adult and young adult. Generally, these populations are more physically active, and smoking behavior is not completely established. Consequently, the dependence level is lower compared with adult smokers, and this may have affected our results. Finally, we argue that the possibility to smoke until 10 a.m. before the experiment session may have reduced the withdrawal symptomatology and the intention to smoke compared with the general smoker population. Future research should consider conducting the study on different age cohorts.

Nonetheless, prescribing physical activity to support smoking cessation has several direct and indirect benefits. First, it might be used in combination with other therapies [7,66,67]). For example, it could be combined with nicotine replacement therapies to improve their efficacy, and help to control desire for a cigarette. Smokers might use exercise as a self-regulation strategy, enhancing their perception of autonomous control during abstinence by improving their self-efficacy. Secondly, it might help smokers and former smokers to monitor weight gain post-cessation. This is of pivotal importance, as weight concerns are a key roadblock in smoking cessation, chiefly in women [68]. Thirdly, as previously reported by other studies, it might help smokers to monitor other health risks, for example, cardiovascular problems and diabetes. Broadly, we argue that exercise might provide greater efficacy if adapted to the individual needs of the smoker, and doing so might foster intrinsic motivation to change smoking behavior.

## 6. Conclusions

This study provides suggestions for the development of more effective interventions based on the introduction of regular physical exercise to support tobacco cigarette smokers. In particular, our results suggested that a key strategy might be the identification of mechanisms that might improve exercise engagement, such as intensity of exercise and participants’ sense of autonomy [69]; the effect of exercise is increased when smokers perceive exercise to be an autonomous choice. In this way, we argue that smokers are able to develop a higher intrinsic motivation to engage with and to maintain physical activity over time. This suggests that, to be useful, the exercise should be tailored according to an individual’s perception of intensity. In fact, finding the intensity of exercise that is able to promote positive affect and relieve negative affect for each smoker may be the better approach to using exercise as an aid to cope with nicotine withdrawal and affect-related distress, helping smokers to quit. Our results on the impact of physical exercise on withdrawal symptoms are coherent with the biological evidence on the association between physical exercise and nicotine responsiveness [11,13]; reported that exercise uses the same “*neurobiological pathway*” of the nicotine in order to reduce craving and withdrawal [11]. In particular, the physical exercise provokes an increase in the β-endorphins in the individual plasma, that contrast with the nicotine withdrawal. An acute and moderate exercise increased in prefrontal-cortex oxygenation, favouring a better inhibitory control and increasing memory and attention skills in polysubstance users [70]. All these mechanisms are central in supporting smokers in their attempts to quit.

It is important that antismoking interventions based on exercise are tailored according to the individual attitudes, preferences and exercise habits of smokers. Overall, this might increase the pleasure, enjoyment and engagement associated with exercise, powering its effectiveness, as it is more probable that people will exercise if it is pleasant and enjoyable. This is in line with the accruing evidence that pleasure and enjoyment are positively correlated with a better adherence to exercise programs, and a higher intrinsic motivation to exercise in general [26].

Furthermore, there are a series of transversal implications of this study that should be considered in the real-world for tobacco cigarette control: firstly, smokers with a low motivation to quit might use the exercise as a strategy to control withdrawal symptoms in the short time, and consequently the number of cigarette smoked per day; secondly the engagement in regular and moderate exercise might bolster better coping strategies to face emotional distress (e.g., frustration, irritability, anger, worry, anxiety), that often activate the desire and urge to smoke; at least, the engagement in moderate physical exercise might increase long-term motivation to adopt a more healthy lifestyle, including smoking interruption.

## Figures and Tables

**Figure 1 healthcare-08-00425-f001:**
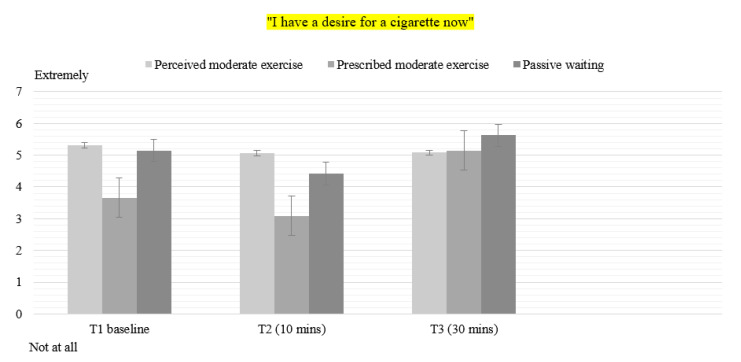
Mean values for item “I have a desire for a cigarette now” in all conditions. T1: Time 1—Baseline pre-intervention. T2: Time 2–10 min of exercise. T3: Time 3–30 min post-intervention.

**Figure 2 healthcare-08-00425-f002:**
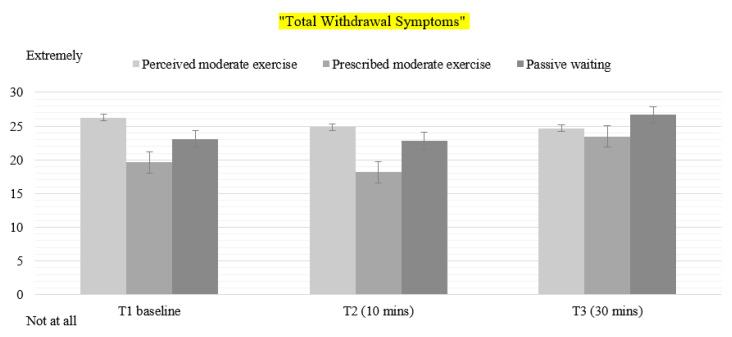
Mean values for Total Withdrawal Symptoms in all conditions. T1: Time 1—Baseline pre-intervention. T2: Time 2–10 min of exercise. T3: Time 3–30 min post-intervention.

**Figure 3 healthcare-08-00425-f003:**
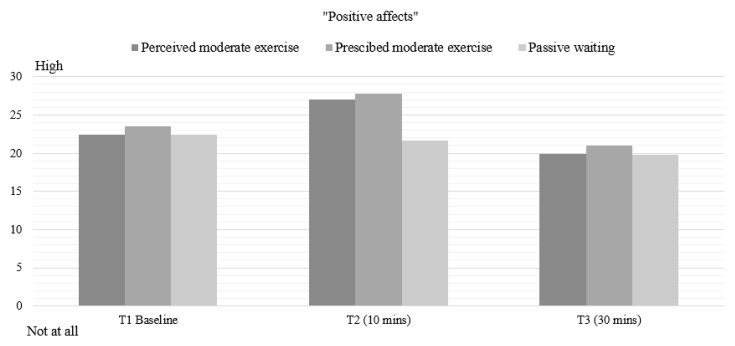
Mean values for Positive effects *(PANAS)* in all conditions. T1: Time 1—Baseline pre-intervention. T2: Time 2–10 min of exercise. T3: Time 3–30 min post-intervention.

**Figure 4 healthcare-08-00425-f004:**
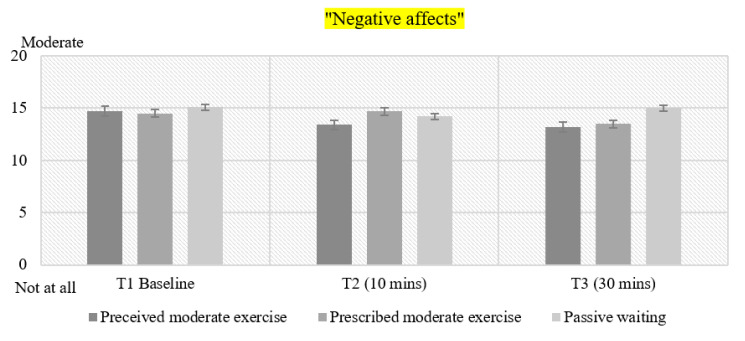
Mean values for Negative affect *(PANAS)* in all conditions. T1: Time 1—Baseline pre-intervention. T2: Time 2–10 min of exercise. T3: Time 3–30 min post-intervention.

**Table 1 healthcare-08-00425-t001:** Mean and standard deviation values for participants’ characteristics: age, numbers of cigarettes per day, numbers of years as smoker, Fagerstrom Test for Nicotine Dependence (FTND), expired carbon monoxide (ECO), and heart rate.

Variables Status		Mean	SD.	N
**Age**	Low-activity	23.92	3.91	26
	Physically Active	23.67	3.26	24
**Male**	Low-activity	-	-	16
	Physically Active	-	-	8
**Female**	Low-activity	-	-	10
	Physically Active	-	-	16
**Number of cigarettes smoked daily**	Low-activity	12.85	3.19	26
	Physically Active	14.17	6.98	24
**Number of years as smoker**	Low-activity	7.31	2.92	26
	Physically Active	5.58	2.21	24
**Timing of the last cigarette before the experiment (h)**	Low-activity	3.15	0.31	26
	Physically Active	3.44	0.7	24
**FTND at baseline**	Low-activity	4.00	1.49	26
	Physically Active	4.00	1.38	24
**Resting Heart Rate (bpm)—Pre-Abstinence**	Low-activity	75.15	10.91	26
	Physically Active	79.58	15.22	24
**Resting Heart Rate (bpm)—Post Abstinence**	Low-activity	75.53	14.37	26
	Physically Active	75.75	6.71	24
**ECO concentration (ppm)—Pre-Abstinence**	Low-activity	2.69	0.618	26
	Physically Active	2.83	0.565	24
**ECO concentration (ppm)—Post Abstinence**	Low-activity	1.31	0.471	26
	Physically Active	1.17	0.381	24

Pre-abstinence: before three hours of abstinence scheduling by the intervention. Post-abstinence: after three hours of abstinence within 30 min of the start of the intervention. FTDN: Fagerstrom Test for Nicotine Dependence. ECO: Expired carbon monoxide. M: Mean. SD: Standard Deviation. N: Number of participants.

**Table 2 healthcare-08-00425-t002:** Mean and Standard deviation for MPAM-R subscales: total motivation toward physical activity, interest, competence, appearance, fitness and social.

MPAM-R Subscales	Status	M	SD	N
Total Motivation toward physical activity	Low-activity	119.23	6.42	26
	Physically Active	147	4.19	24
Interest	Low-activity	30.15	9.38	26
	Physically Active	38.25	6.16	24
Competence	Low-activity	26.62	11.28	26
	Physically Active	32.50	7.52	24
Appearance	Low-activity	21.92	9.05	26
	Physically Active	29.17	9.56	24
Fitness	Low-activity	23.46	6.25	26
	Physically Active	28.33	4.57	24
Social	Low-activity	14.77	8.35	26
	Physically Active	16.50	6.73	24

M: Mean. D: Standard Deviation. N: Number of participants.

**Table 3 healthcare-08-00425-t003:** Mean and standard deviation values for rating of perceived exertion and heart rate in all conditions.

Status	Time	Condition	*Rating of Perceived Exertion*		*Heart Rate*	
			M	SD	M	SD
Low Active		PER	4 (1.3)	1.13	102.230	15.73
	T1 (2.5 min)	OBJ	5.53	1.63	142.31	6.07
		SED	0.61	0.38	68.76	12.33
		PER	4.53	1.52	104.07	19.24
	T2 (7.5 min)	OBJ	5.92	1.89	140.15	5.15
		SED	0.53	0.64	69.53	13.60
Physically Active		PER	3.33	0.96	129	22.44
	T1 (2.5 min)	OBJ	4.58	1.58	145.25	8.40
		SED	0.167	0.38	74.33	7.62
		PER	3.5	0.97	131.08	25.46
	T2 (7.5 min)	OBJ	5.167	1.43	146.33	9.01
		SED	0.25	0.44	73.25	9.27

M: Mean. SD: Standard Deviation. N: Number of participants.

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
