# Peer review of "Short Bouts of Physical Activity Are Associated with Reduced Smoking Withdrawal Symptoms, But Perceptions of Intensity May Be the Key"

_healthcare, 2020, doi:10.3390/healthcare8040425_

Round 1

Reviewer 1 Report

The manuscript by Masiero et al. investigated whether a short bout of moderate exercise as measured by bicycle ergometer reduced the desire to smoke in heavy smokers. They found that a 10-min stationary cycling could indeed decrease the desire to smoke during the exercise, compared to controls who were reading magazines during the same time frame, but regained the desire 30 mins after the exercise. The authors interpreted these results as the anti-depression effects of exercise that biologically reduce the responsiveness of nicotine receptor, rather than as a distraction effect. This is a well-written manuscript with a good statistical description and analysis, but the research design may need to be improved to provide more convincing evidence for the conclusion currently drawn in the manuscript. My comments are as follows.     

1. To exclude a distraction effect as a possible reason for their observation, the authors may need to include a control group who were taking 10 min’s mental exercise of moderate intensity, such as typing a 3 pages long paper or answering a quiz, for a fair comparison.    

2. To strengthen their hypothesis for an effect of exercise on nicotine abstinence, the authors may need to discuss more about the underlying biological and/or molecular mechanism, for example, the connection between exercise-upregulated endorphin level and nicotine responsiveness, etc. It would be helpful if the authors could revise their Conclusion section in this regard.

3. The emerging adult and young adult smokers tested in this study are far less representative of the heavy smokers in the real world who are mainly middle-aged people. The latter people have a very high risk of developing cardiovascular diseases and cancers as a result of systemic impairment of the immune system by the tobacco components other than nicotine. These people need real help. In this regard, the present study is heavily limited by a selection bias.

4. It would be helpful if the authors could add the labels of the Y-axis of all the figures in the manuscript.

Author Response

REVIEWER 1 

The manuscript by Masiero et al. investigated whether a short bout of moderate exercise as measured by bicycle ergometer reduced the desire to smoke in heavy smokers. They found that a 10-min stationary cycling could indeed decrease the desire to smoke during the exercise, compared to controls who were reading magazines during the same time frame, but regained the desire 30 mins after the exercise. The authors interpreted these results as the anti-depression effects of exercise that biologically reduce the responsiveness of nicotine receptor, rather than as a distraction effect. This is a well-written manuscript with a good statistical description and analysis, but the research design may need to be improved to provide more convincing evidence for the conclusion currently drawn in the manuscript. My comments are as follows.  

Thanks to the reviewer for your pivotal comments that they have helped us to improve our manuscript. We provided a point-by-point response.

We have worked through the paper, incorporating suggestions. Modifications in the manuscript have been highlighted in yellow. 

Please see the attachment file with the manuscript revised.

P1: 1. To exclude a distraction effect as a possible reason for their observation, the authors may need to include a control group who were taking 10 min’s mental exercise of moderate intensity, such as typing a 3 pages long paper or answering a quiz, for a fair comparison.    

Thanks for the suggestion to add another control group in order to monitor a possible distraction effect. In the current study design, we have used a “passive waiting condition” as a control group, in which each participant performed some activities such as read newspapers. However, we did not include in this condition mental exercise of moderate intensity such as, answering a quiz, and so on. For this reason, we add a sentence in the limitations paragraph to recognize this point.

Page 12 - Line [432-436]

P2: 2. To strengthen their hypothesis for an effect of exercise on nicotine abstinence, the authors may need to discuss more about the underlying biological and/or molecular mechanism, for example, the connection between exercise-upregulated endorphin level and nicotine responsiveness, etc. It would be helpful if the authors could revise their Conclusion section in this regard.

We added in the Conclusion a specific section on biological and molecular aspects as reinforcement to our observation.

Page 13 - Line [467-474]

R1: 3. The emerging adult and young adult smokers tested in this study are far less representative of the heavy smokers in the real world who are mainly middle-aged people. The latter people have a very high risk of developing cardiovascular diseases and cancers as a result of systemic impairment of the immune system by the tobacco components other than nicotine. These people need real help. In this regard, the present study is heavily limited by selection bias.

We agree with the reviewer. We discuss this in the limitation section.

Page 12 - Line [426-429]

P4: It would be helpful if the authors could add the labels of the Y-axis of all the figures in the manuscript.

This point has been fixed.

Reviewer 2 Report

Thank you for the opportunity to review this paper, which initially (via the title and abstract) promised to be of reasonable interest to those helping smokers to quit. However, the current study design and data analysis, according to the current presentation, are flawed in several important ways that would not warrant my recommendation for publication.

Some examples of such concerns/flaws are:

  1. In the Abstract, the distinction between low-active and physically active smokers was not mentioned. This implies its irrelevance, despite it taking 'centre stage' in the main text.
  2. The comparison of the OBJ & PER groups is flawed, not knowing the potential overlap between them. Therefore, any subsequent comparison/analysis is rendered meaningless.
  3. Only 5 minutes to move from 1 condition to the next, which did not allow adequate 'wash over' period, hence contamination of effects (if any) was highly likely.
  4. Each condition lasted 30 minutes, during which several measures were taken, raising doubt over the legitimacy/accuracy/credibility of each measure taken within such a short period of time.
  5. The duration of these conditions and the outcome measures is so miniscule and irrelevant/meaningless for real-life utility (for smokers wanting/needing to quit - how great is an achievement if one reports positive mood and not craving for a cigarette during 30 minutes??!!). This flaw is amplified, given that the inclusion criterion of >=10 cigarettes smoked per day, which is very mild in real life.
  6. I am not convinced that the power and sample size analysis was correct or adequate. The effect size desired, or used in the power calculation, was not stated, especially when there were many outcomes of interest - the authors would be expected to state for which outcome(s) the power analysis was performed. If it was performed for all the outcomes, one would expect to see a range of sample size (min and max) required, depending on the outcomes considered.
  7. If I misunderstood any of the above points, I am sorry for that, but that would also demonstrate how suboptimally the study was presented.
  8. Table 1 showed age, but not gender. 
  9. Why select such a restricted age range 18-35? Sampling method was also severely susceptible to bias. 

Author Response

POINT 1: Thank you for the opportunity to review this paper, which initially (via the title and abstract) promised to be of reasonable interest to those helping smokers to quit. However, the current study design and data analysis, according to the current presentation, are flawed in several important ways that would not warrant my recommendation for publication.

RESPONSE 1: Although we respect the comments by the reviewer, we do not believe that the current study design not the data analysis is in any way flawed. We hope that the detailed justification and clarifications provided below would address the concerns of the reviewer.

We have worked through the paper, incorporating suggestions. Modifications in the manuscript have been highlighted in yellow. Please see the revised manuscript attached.

POINT 2: Some examples of such concerns/flaws are: In the Abstract, the distinction between low-active and physically active smokers was not mentioned. This implies its irrelevance, despite it taking 'centre stage' in the main text.

RESPONSE 3:We desire to highlight that the main aim of the study was twofold and not exclusively based on the distinction between physically active and low-active smokers: the primary aim, to assess the effectiveness of a short bout (10 minutes) of moderate intensity exercise to reduce withdrawal symptomatology, craving and negative affect; the secondary aim was to assess how the effectiveness of a short bout of moderate exercise can be modulated by the perception of intensity in physically active and low-active smokers. Notwithstanding, we added a sentence on low-active and physically active smokers in the abstract in order to clarify this point.

Page 1 - Line [18-21]

POINT 3: The comparison of the OBJ & PER groups is flawed, not knowing the potential overlap between them. Therefore, any subsequent comparison/analysis is rendered meaningless.

RESPONSE 3: We apologize, but are not sure we understand the reviewers comment regarding “potential overlap”. a. The categorization used in the current study between prescribed objective moderate intensity (respectively OBJ) and perceived moderate-intensity exercise (respectively PER) has based on two different methodologies that have a good internal and external consistency and validity. Furthermore, these measures are typically used to classify the strength and the physical involvement of the participants in the exercise/activity. Respectively, the OBJ condition has been determinate according to Karvonen method (55% heart rate reserve), while in PER condition each participant established what they perceived to be moderate-intensity exercise according to Borg Rating of Perceived Exertion RPE level. Comparison between OBJ and PER groups has also been reported in well-cited publications as a valid methodology that has good internal and external consistency and validity. Please see the following references in which are discussed some studies that have used a similar approach, also in different domains.

  • Rose, E. A., & Parfitt, G. (2012). Exercise experience influences affective and motivational outcomes of prescribed and self‐selected intensity exercise. Scandinavian Journal of Medicine & Science in Sports22(2), 265-277.
  • Ekkekakis, P., & Lind, E. (2006). Exercise does not feel the same when you are overweight: the impact of self-selected and imposed intensity on affect and exertion. International journal of obesity30(4), 652-660.
  • Zourbanos, N., Hatzigeorgiadis, A., Tsiami, A., Tzatzaki, T., Georgakouli, K., Manthou, E., ... & Hassandra, M. (2016). An initial investigation of smokers’ urges to smoke and their exercise intensity preference: A mixed-methods approach. Cogent Medicine, 3(1), 1149043.

POINT 4: Only 5 minutes to move from 1 condition to the next, which did not allow adequate 'wash over' period, hence contamination of effects (if any) was highly likely.

RESPONSE 4: Apologies for the typo but the time for the rest between conditions was 15 minutes and not 5 minutes. Generally, 15-20 minutes may be considered an adequate time to eliminate the effect of the previous condition/intervention (OBS, SED, PW) and had been used consistently in the literature as a valid wash out time frame, for example, Taylor, A. H., Katomeri, M., Ussher, M. (2005). Acute effects of self-placed walking on urges to smoke during temporary smoking abstinence. Psychopharmacology, 181(1), 1-7. Furthermore, in our study, heart rate was monitored in each participant at different time points partly in order to monitor that each participant came back to a normal physical condition after the condition/intervention, following a 15 min washout period. Our data thus revealed that 15 mins was an adequate containment strategy to avoid contamination effects.

POINT 5: Each condition lasted 30 minutes, during which several measures were taken, raising doubt over the legitimacy/accuracy/credibility of each measure taken within such a short period of time.

RESPONSE 5: We understand your concerns about the time. However, the measures used in the current study are easy to fill for the participants and not required a long period to fill them. Furthermore, before the experiment session each participant has received a debriefing in which measures have been presented and explained in detailed way by a trained researcher. In this way at each time point each participant know what measures have to fill and in which way should be filled (others studies have used similar design such as Everson, E. S., Daley, A. J., & Ussher, M. (2008). The effects of moderate and vigorous exercise on desire to smoke, withdrawal symptoms and mood in abstaining young adult smokers. Mental Health and Physical Activity1(1), 26-31; Ussher, M., Nunziata, P., Cropley, M., & West, R. (2001). Effect of a short bout of exercise on tobacco withdrawal symptoms and desire to smoke. Psychopharmacology158(1), 66-72).

POINT 6: The duration of these conditions and the outcome measures is so miniscule and irrelevant/meaningless for real-life utility (for smokers wanting/needing to quit - how great is an achievement if one reports positive mood and not craving for a cigarette during 30 minutes??!!). This flaw is amplified, given that the inclusion criterion of >=10 cigarettes smoked per day, which is very mild in real life.

RESPONSE 6: The aim of the study was to assess the effectiveness of a short bout of moderate exercise to modulate craving for cigarette and mood in a short period of time (30 minutes), and not to assess the long-term consequences on craving and mood.

In order to achieve this specific aim, we developed our study design coherently with other studies conducted on this specific issue, for example, we reported the following references:

  • Daniel, J., Cropley, M., Ussher, M., & West, R. (2004). Acute effects of a short bout of moderate versus light intensity exercise versus inactivity on tobacco withdrawal symptoms in sedentary smokers. Psychopharmacology174(3), 320-326;
  • Ussher, M., West, R., Doshi, R., & Sampuran, A. K. (2006). Acute effect of isometric exercise on desire to smoke and tobacco withdrawal symptoms. Human Psychopharmacology: Clinical and Experimental21(1), 39-46;
  • Scerbo, F., Faulkner, G., Taylor, A., & Thomas, S. (2010). Effects of exercise on cravings to smoke: The role of exercise intensity and cortisol. Journal of Sports Sciences28(1), 11-19).

Overall, we were interested to gather mechanistic evidence over its efficacy. Based on our current evidence it would be intriguing to assess whether exercising repeatedly over a longer period of time would enable a prolonged effect of craving and mood. Furthermore, concerning the cut-off used to assess daily cigarette per day we referred to the international literature of tobacco smoking behaviour, nicotine addiction and WHO classification suggested that to smoke more than 10 cigarettes per day is a valid inclusion criterion, and it is extensively used in the research filed on smoking behaviour as inclusion criteria.

For example, Bullen and colleagues (2013) used in their randomized controlled trial to assess the usability/effectiveness of the e-cigarette as smoking cessation aid the same inclusion criteria (Bullen, C., Howe, C., Laugesen, M., McRobbie, H., Parag, V., Williman, J., & Walker, N. (2013). Electronic cigarettes for smoking cessation: a randomised controlled trial. The Lancet, 382(9905), 1629-1637). Moreover, several measures used to assess nicotine addiction, such as for example Fagerstrӧm Test of Nicotine Dependence, used 10 cigarettes per day as a starting cut-off.

Also, many Authors such as Benowitz (2010) stated that light smokers are individuals that smoke ≤5 cigarettes per day (Benowitz, N.L. (2010). Nicotine addiction. New England Journal of Medicine, 362(24), 2295-2303). Notwithstanding we want to underline that the force of the nicotine addiction and consequently the withdrawal symptoms are not exclusively related to the number of cigarettes per day, by also to the individual metabolism processes of the nicotine. Although withdrawal symptoms may not be prominent, many light and occasional smokers have difficulty quitting. Some of them have a high level of dependence, but with pharmacodynamics that differ from those in heavier smokers.

POINT 7: I am not convinced that the power and sample size analysis was correct or adequate. The effect size desired, or used in the power calculation, was not stated, especially when there were many outcomes of interest - the authors would be expected to state for which outcome(s) the power analysis was performed. If it was performed for all the outcomes, one would expect to see a range of sample size (min and max) required, depending on the outcomes considered.

RESPONSE 7: The sample size has been calculated in line with previous studies conducted in this area (Taylor, A., Katomeri, M., & Ussher, M. (2005). Acute effects of self-paced walking on urges to smoke during temporary smoking abstinence. Psychopharmacology, 181, 1-7.; Taylor, A., Katomeri, M., & Ussher, M. (2006). Effects of walking on cigarette cravings and affect in the context of Nesbitt’s paradox and the circumplex model. J Sport Exercise Psy, 28, 18-31) and a power analysis was conducted as required (using the following software: G*Power Version 3.1). The sample size was established based on the study designs aim which is to assess the effectiveness of a short bout of moderate intensity exercise to reduce withdrawal symptomatology (primary) and to assess how the effectiveness of a short bout of moderate exercise can be modulated by the perception of intensity in physically active and low-active smokers (secondary). According to this, a sample size of 15 has been considered adequate for a moderate to high effect size, with a power (1 – β) of 0.8, and an α-value of 0.05. A priori power analysis (determined that a sample size of 18 for both the SED and PA groups) would be sufficient to achieve power of 0.8 and an α-value of 0.05.

POINT 8: If I misunderstood any of the above points, I am sorry for that, but that would also demonstrate how suboptimally the study was presented.

RESPONSE 8: We hope that our explanations help you to clarify doubts and misunderstanding.

POINT 9: Table 1 showed age, but not gender. 

RESPONSE 9: We added this missing information in Table 1.

POINT 10: Why select such a restricted age range 18-35? Sampling method was also severely susceptible to bias. 

RESPONSE 10: We have used a snowball sampling and participants have been enrolled at University of Surrey. We understand the risk of selection bias. Thus, we have added a detailed explanation about the age range and selection bias in the limitation of the study.

Page 12 - Line [426-429]

Reviewer 3 Report

This is a well-prepared manuscript aimed at significant public health problems. This study is novel and the methods applied in the study are correct. This study was conducted in a laboratory setting and pose a basis for further research on this topic.

There are some minor comments:

1) Abstract section - please clearly define the aim of the study - the current version (first line of the abstract section) is unclear and does not refer to smoking.

2) Methods - please clearly define sampling methods/recruitment strategies. Where the study was carried out? Who was invited to take part? How about the recruitment strategy? How the recruitment strategy affected results? 

3) Please clearly define questions used for the assessment of smoking status. Do the authors use questions recommended by the WHO (smoking at least 100 tobacco products; and currently tobacco use)

4) Please provide 2-3 sentences related to the "practical implications" of this study

5) Limitations section should be placed as the last paragraph of the discussion section (before the conclusions section)

6) The first sentence of the conclusions section (line 392) should be re-written. Please provide conclusions that are based on your own findings (and add some sentences about further need), without overwhelming conclusions

Author Response

This is a well-prepared manuscript aimed at significant public health problems. This study is novel and the methods applied in the study are correct. This study was conducted in a laboratory setting and pose a basis for further research on this topic.

We thank the reviewer for their helpful comments and critical appraisal of the manuscript. We have integrated your comments and suggestions. Modifications in the manuscript have been highlighted in yellow. Please see the revised manuscript attached.

There are some minor comments:

POINT 1: 1) Abstract section - please clearly define the aim of the study - the current version (first line of the abstract section) is unclear and does not refer to smoking.

We fixed this point and reformulating better the sentence trying to stress the primary of the study.

Page 1 - Line [18-22]

POINT 2: 2) Methods - please clearly define sampling methods/recruitment strategies. Where the study was carried out? Who was invited to take part? How about the recruitment strategy? How the recruitment strategy affected results? 

RESPONSE 2: We fixed this point. We added in the method section a detailed description of the recruitment strategies.

Page 4 - Line [149-152]

POINT 3: 3) Please clearly define questions used for the assessment of smoking status. Do the authors use questions recommended by the WHO (smoking at least 100 tobacco products; and currently tobacco use)

RESPONSE 3: The smoking status has been assessed using a theoretical framework WHOs’ definition. Consistently to this, we defined as a tobacco cigarette smoker: “who smoke at least 10 cigarettes a day for at least two years”. We added inclusion and exclusion criteria in the procedure paragraph in order to clarify this point, and other related points.

Page 4 - Line [152-157]

POINT 4: Please provide 2-3 sentences related to the "practical implications" of this study

RESPONSE 4: We added this to the conclusion.

Page 13 – Line [48-488]

POINT 5: 5) Limitations section should be placed as the last paragraph of the discussion section (before the conclusions section)

RESPONSE 5: We fixed this point

Page 12 - Line [424]

POINT 6: 6) The first sentence of the conclusions section (line 392) should be re-written. Please provide conclusions that are based on your own findings (and add some sentences about further need), without overwhelming conclusions.

RESPONSE 6: The conclusions were edited coherently to the received suggestions

Page 12-13 - Line [457-460]

Page 12-13 - Line [467-474]

Page 12-13 - Line [481-488]

Round 2

Reviewer 1 Report

The authors have mostly addressed the questions raised by this reviewer.